# Gene Expression Profiling and Protein Analysis Reveal Suppression of the C-Myc Oncogene and Inhibition JAK/STAT and PI3K/AKT/mTOR Signaling by Thymoquinone in Acute Myeloid Leukemia Cells

**DOI:** 10.3390/ph15030307

**Published:** 2022-03-03

**Authors:** Belal Almajali, Muhammad Farid Johan, Abdullah Saleh Al-Wajeeh, Wan Rohani Wan Taib, Imilia Ismail, Maysa Alhawamdeh, Nafe M. Al-Tawarah, Wisam Nabeel Ibrahim, Futoon Abedrabbu Al-Rawashde, Hamid Ali Nagi Al-Jamal

**Affiliations:** 1School of Biomedicine, Faculty of Health Sciences, Universiti Sultan Zainal Abidin (UniSZA), Kuala Nerus 21300, Terengganu, Malaysia; bel_basss@yahoo.com (B.A.); wanrohani@unisza.edu.my (W.R.W.T.); imilia@unisza.edu.my (I.I.); futoonrawashdeh1001@gmail.com (F.A.A.-R.); 2Department of Haematology, School of Medical Sciences, Universiti Sains Malaysia, Kubang Kerian 16150, Kelatan, Malaysia; faridjohan@usm.my; 3Anti-Doping Lab Qatar, Doha 27775, Qatar; asaleh@adlqatar.qa; 4Department of Medical Laboratory Sciences, Faculty of Sciences, Mutah University, Alkarak 61710, Jordan; maysa5005@yahoo.com (M.A.); nafitawa77@gmail.com (N.M.A.-T.); 5Department of Biomedical Sciences, College of Health Sciences, QU Health, Qatar University, Doha 122104, Qatar; w.ibrahim@qu.edu.qa

**Keywords:** thymoquinone, leukemia, c-Myc, JAK/STAT, PI3K/AKT, signaling, apoptosis

## Abstract

Overexpression of c-Myc plays an essential role in leukemogenesis and drug resistance, making c-Myc an attractive target for cancer therapy. However, targeting c-Myc directly is impossible, and c-Myc upstream regulator pathways could be targeted instead. This study investigated the effects of thymoquinone (TQ), a bioactive constituent in *Nigella sativa*, on the activation of upstream regulators of c-Myc: the JAK/STAT and PI3K/AKT/mTOR pathways in HL60 leukemia cells. Next-generation sequencing (NGS) was performed for gene expression profiling after TQ treatment. The expression of *c-Myc* and genes involved in JAK/STAT and PI3K/AKT/mTOR were validated by quantitative reverse transcription PCR (RT-qPCR). In addition, Jess assay analysis was performed to determine TQ’s effects on JAK/STAT and PI3K/AKT signaling and c-Myc protein expression. The results showed 114 significant differentially expressed genes after TQ treatment (*p* < 0.002). DAVID analysis revealed that most of these genes’ effect was on apoptosis and proliferation. There was downregulation of *c-Myc*, PI3K, AKT, mTOR, JAK2, STAT3, STAT5a, and STAT5b. Protein analysis showed that TQ also inhibited JAK/STAT and PI3K/AKT signaling, resulting in inhibition of c-Myc protein expression. In conclusion, the findings suggest that TQ potentially inhibits proliferation and induces apoptosis in HL60 leukemia cells by downregulation of c-Myc expression through inhibition of the JAK/STAT and PI3K/AKT signaling pathways.

## 1. Introduction

Acute myeloid leukemia (AML) is the most common acute leukemia in adults [1]. AML is a hematological malignancy characterized by multiple acquired mutations that affect cell biological processes such as cell growth, proliferation, and apoptosis [2]. The *c-Myc* oncogene is overexpressed in HL60 AML cells [3]. It mediates multiple tumor cell survival pathways in most human cancers and represents a promising therapeutic target in several cancers [4,5]. However, drugs cannot directly target c-Myc because it has no specific active site for small molecules. It is found mainly in the nucleus, and it is impossible to target the nuclear c-Myc by monoclonal antibodies [6]. Therefore, targeting the upstream signaling of c-Myc and the PI3K/AKT/mTOR and JAK/STAT pathways could indirectly inhibit c-Myc expression [7]. In mammals, the JAK/STAT and PI3K/AKT/mTOR pathways are the principal signaling mechanisms for many growth factors and cytokines, and the activation of these pathways stimulates cell proliferation, differentiation, cell migration, and apoptosis [8]. However, the JAK/STAT and PI3K/AKT/mTOR pathways are constitutively activated in many cancers, including leukemia, due to the fact of genetic events such as mutations in cytokine receptors and aberrant chromosomal translocations [9]. C-Myc was overexpressed by constitutive activation of the JAK/STAT and PI3k/AKT signaling pathways [10].

Advances in AML treatment have led to a better prognosis for younger patients, but the outcomes were poor for elderly patients, who make up the majority of new cases [11]. Therefore, there is an urgent need for a novel and more efficient treatment for such malignancies. Accumulated evidence based on in vivo and in vitro studies indicated the anti-cancer properties of compounds obtained from natural products [12]. Thymoquinone (TQ), chemically known as 2-methyl-5-isopropyl-1,4-benzoquinone, is an active phytochemical compound of *Nigella sativa* (black seeds) that has shown anti-proliferative, pro-apoptotic, and anti-metastatic properties on numerous cancer cell types including leukemia cells [13,14]. In the present study, to understand the effect of TQ on gene regulation in AML cells, gene expression profiling using next-generation sequencing (NGS) technology was performed to determine the changes in gene expression after TQ treatment. The Database for Annotation, Visualization, and Integrated Discovery (DAVID) online bioinformatics tool was used to evaluate the changes in mRNA expression in the HL60 leukemia cells after TQ treatment. NGS-based analyses have produced significant new understandings into the molecular pathogenesis of AML [15,16]. The expression of targeted genes involved in JAK/STAT and PI3K/AKT/mTOR was verified by RT-qPCR in TQ-treated HL60 cells compared to untreated cells. Moreover, the protein expression and their phosphorylation status in the JAK/STAT and PI3K/AKT/mTOR pathways and the c-Myc protein were studied using Jess simple Western analysis. Therefore, the current study improved the present knowledge of TQ effect mechanisms and provides a practical technique for future studies on pharmacogenomics.

## 2. Materials and Methods

### 2.1. Cell Line and Growth Media

The HL60 cell line was obtained from (Elabscience Biotech Co., Ltd., Wuhan, China) and cultured in a complete medium containing Roswell Park Memorial Institute (RPMI) 1640 Medium (Nacalai Tesque, Kyoto, Japan), fetal bovine serum (FBS) 10% (Tico Europe, Holland), and 1% penicillin/streptomycin (Nacalai Tesque, Kyoto, Japan) in T-25 culture flasks at 37 °C with 5% CO_2_ in a humid incubator [17]. The medium was changed every 3–4 days. The HL60 cells were passaged at 80–90% confluence for the next experiments.

### 2.2. TQ Treatment

TQ was purchased from Sigma–Aldrich (Munich, Germany). Then, 5 mM of TQ was prepared in dimethyl sulfoxide (DMSO) as a stock solution. Based on our published data, cells were treated with 2 µM of TQ diluted in a complete RPMI culture medium as half-maximal inhibitory concentration (IC_50_) and incubated for 48 h [14].

### 2.3. RNA Extraction

According to the manufacturer’s protocol, total RNA from TQ-treated and untreated HL60 cells were extracted using the ReliaPrep™ RNA Cell Miniprep System (Promega, Madison, WI, USA). The concentration and purity of RNA were measured using a NanoPhotometer^®^ NP80 (Implen GmbH, München, Germany).

### 2.4. Next-Generation Sequencing for mRNA Expression Profiling

The gene expression profiles of mRNAs were conducted using the NGS analysis, Illumina platform, Agilent Technologies (Seoul, South Korea). The quality of sequencing raw data was inspected using FastQC (version 0.11.8). Low-quality reads (i.e., reads with a Phred score less than Q20), adapter, and poly G sequences were removed using Fastp [18]. The clean reads were inspected again using FastQC to make sure the unwanted reads were removed. The clean reads were then aligned against a human reference genome (version 38, downloaded from the Ensembl database) using Spliced Transcripts Alignment to a Reference (STAR) version 2.7.3a [19]. STAR is an aligner developed to manage the obstacles of RNA-seq data mapping using a method to account for spliced alignments. Prior to running the alignment, the reference genome index was built, and exon information was extracted from the latest version of the gene annotation file (version 35, downloaded from GENCODE) using STAR. Next, the aligned RNA-seq reads, in BAM format, were quantified using featureCounts (version 1.6) [20]. This gene-level quantification system uses a gene transfer format (GTF v35) file with gene models and counts the number of reads that align with each gene (i.e., read count).

In this study, DESeq2 [21] was adopted for differential gene expression (DEG) analysis. The tool takes featureCounts output (raw counts) as input. Many factors influence raw read counts such as transcript length and the total number of reads. Thus, to compare expression levels between samples, raw read normalization was performed. DESeq2 conducts internal normalization, where the geometric mean is calculated per gene across all samples. Prior to running the DEG analysis, sample-level and gene-level QC were performed on the count data. At the sample level QC, principal component analysis (PCA) was constructed to identify any potential outliers. At gene-level QC, genes that had zero or few mean read counts (less than 10) were omitted for downstream analysis. This increases the power to detect differentially expressed genes. Lastly, DESeq2 fits negative binomial generalized linear models for genes and utilizes the Wald test for significance testing. DEGs that passed the following filters were categorized as significant DEGs: *p*-adjusted value (*p-adj*) < 0.05 and log fold change (lfc) > 1 or <−1, which is equivalent to >2- or <−2-fold change, respectively. All significantly affected genes were uploaded into the DAVID database to analyze functions and pathways associated with TQ treatment.

### 2.5. DAVID Database Analysis

The Database for Annotation, Visualization, and Integrated Discovery (DAVID) is a free powerful software that classifies functional genes (https://david.ncifcrf.gov/) (accessed on 5 October 2021) [22]. It combines biological processes, the Kyoto Encyclopedia of Genes and Genomes (KEGG) pathway, and Gene Ontology (GO). By using the software, genes affected after TQ treatment can be grouped to sets of related biological functions or signaling pathways by calculating the similarity of global annotation profiles with the agglomeration algorithm method. This study selected the expression analysis systematic explorer (EASE) score = 0.1 as the default and specified pathways with a Bonferroni *p*-value of <0.05 as significant.

### 2.6. Reverse Transcription Quantitative Polymerase Chain Reaction (RT-qPCR)

cDNA was synthesized from RNA (100 ng) that was extracted from treated and untreated HL60 cells using the GoTaq 2-Step RT-qPCR System (Promega, USA) according to the manufacturer’s protocol. The amplifications of PCR were performed using 50 µL of GoTaq PCR Master Mix and 2 µL of cDNA: a denaturing step at 95 °C for 2 min followed by 40 cycles of denaturation at 96 °C for 15 s, then annealing and an extension step at 60 °C for 1 min using a StepOne RT-qPCR System (Applied Biosystems, Waltham, MA, USA). StepOne Software v2.3 (Applied Biosystems) was used in data analysis. Beta-actin (*β-actin*) was used as a housekeeping gene. The fold changes of gene expression levels were evaluated by relative quantification of targeted genes using the 2^−ΔΔCq^ method [23]. The primer sequences are listed in Table 1. All experiments were performed in triplicate.

### 2.7. Protein Analysis Using Jess Simple Western Analysis

Treated and untreated HL60 leukemia cells were harvested and washed with ice-cold PBS. The protein lysate was obtained by incubating the cells with RIPA buffer containing protease inhibitor cocktail (Nacalai Tesque, Kyoto, Japan) for 15 min on ice. Then, the lysate was centrifuged at 10,000× *g* for 10 min at 4 °C. The BCA Protein Assay Kit (Heart, Xi’an, China) was used to measure the concentrations of the proteins in the supernatant. Capillary Western analyses were performed using the Jess simple Western analysis (Neoscience, Selangor, Malaysia) according to the manufacturer’s protocol. Briefly, cell lysates were diluted with 0.1X sample buffer to a concentration of 1 mg/mL and then diluted with fluorescent 5X master mix at a 4:1 ratio and heated at 95 °C for 5 min. In total, 3 μL of each sample were loaded into the plate. A 12–230 kDa cartridge was used. The target proteins were immune-probed with primary antibodies that were diluted in antibody buffer at a 1:10 ratio (i.e., anti-JAK, anti-pJAK, and anti-c-Myc (Novus Biological, Littleton, CO, USA); anti-STAT3, anti-pSTAT3, anti-STAT5, anti-pSTAT5, anti-AKT, anti-pAKT, anti-PTEN, and anti-PI3K (R&D Systems, Minneapolis, MN, USA); anti-pPI3K (Tyr458/Tyr199) (Invitrogen, Waltham, MA, USA)) followed by HRP-conjugated secondary antibodies. In total, 10 μL of primary and secondary were loaded for each sample. The plate was spun down for 5 min at 1000× *g* to remove bubbles. Subsequently, capillaries and the plate were loaded into the Jess machine. Protein separation, blocking, antibody incubation, and signal detection were conducted automatically by the Jess system. The Compass software provided by the manufacturer was used in the data analysis. The normalization reagent detects proteins in capillaries by binding a biomolecule to an amine group and removing the housekeeping protein that can cause unreliable expression. The Jess technology did not need a control in the experiment.

### 2.8. Statistical Analysis

Kruskal–Wallis and Mann–Whitney tests were conducted for statistical analysis using GraphPad Prism 8.4.3 (San Diego, CA, USA), and *p* < 0.05 was considered as significant.

## 3. Results

### 3.1. Thymoquinone Induced Differentially Expressed Genes in HL60 Cells

The gene expression profiling of HL60 leukemia cells before and after TQ-treatment was performed for the whole transcriptome using NGS analysis. There were more than 1900 genes that showed different expression in TQ-treated HL60 leukemia cells. Volcano plots of the gene expression data in HL60 associated with TQ treatment (Figure 1) show significant DEGs at *p-adj* < 0.05 and fold change with absolute value >1. There were 54 genes that showed significant upregulation, and 60 genes were significantly downregulated after treatment of HL60 cells with 2 μM TQ. In general, tumor suppressor genes, including pro-apoptotic genes, were upregulated, while many oncogenes were downregulated. The gene expression profiling analysis also showed 100 obviously differentially expressed genes of down- versus upregulated genes in three TQ-treated HL60 cell samples compared to three untreated samples (Figure 2).

Free online DAVID Functional Annotation Bioinformatics Microarray Analysis was conducted to analyze possible mechanisms of the top modulated genes in HL60 cells related to TQ treatment. The biological process analysis (Figure 3) showed that the majority of genes were distributed in G protein-coupled receptor pathways (11 genes were involved), immune response (10 genes), regulation of GTPase activity (9 genes), inflammatory response (8 genes), cytokine-mediated and cell surface receptor pathways (7 genes each), regulation of angiogenesis and cell adhesion (6 genes each), and regulation of apoptosis (5 genes). In cellular components (Figure 4). The plasma membrane was the active site for the majority of affected genes (86 genes).

Analyzing up- and downregulated genes separately showed their critical roles in specific biological processes (Table 2 and Table 3).

DAVID analysis of the NGS results showed that the differentially expressed genes were primarily involved in signaling pathways, proliferation, angiogenesis, and apoptosis processes in leukemia cells. Some of these genes control the activation of the JAK/STAT and PI3K/AKT/mTOR signaling such as CCL2, CXCR2, and PRDM8.

### 3.2. TQ Downregulated the Expression of JAK/STAT Pathways Genes

RT-qPCR was used to confirm the expression changes of *JAK2*, *STAT3*, *STAT5a*, and *STAT5b* genes in HL60 cells after treatment with TQ (Figure 5). It was found that *JAK2*, *STAT3*, *STAT5a*, and *STAT5b* gene expressions were significantly inhibited in TQ-treated cells (*p* < 0.001) compared with untreated cells as shown in Table 4. The NGS results also indicated a significant downregulation in targeted genes.

### 3.3. TQ Downregulated the Expression of the PI3K/AKT Pathway Genes

The results of RT-qPCR revealed significant downregulation of *AKT* (*p* < 0.001), *PI3K* (*p* < 0.001)*,* and *mTOR* (*p* < 0.001) expression in TQ-treated HL60 cells as shown in Table 5 and Figure 6. These results were consistent with the NGS results.

### 3.4. TQ Downregulated the Expression of the c-Myc Gene

This study aimed primarily to study the effect of TQ on c-Myc. Therefore, the effect of TQ on the c-Myc mRNA level was evaluated in HL60 cells by RT-qPCR (Figure 7). The results showed that TQ significantly decreased the expression of *c-Myc* (*p* < 0.001) (Table 6). The NGS and RT-qPCR both indicated downregulation of *c-Myc* expression in TQ-treated cells.

According to the results of the current study, TQ reduced the gene expression of *c-Myc* by inhibiting the expression of JAK/STAT and PI3K/AKT/mTOR genes including *JAK2*, *STAT3*, *STAT5a*, *STAT5b*, *PI3K*, *AKT*, and *mTOR*.

### 3.5. TQ Inhibited JAK/STAT Signaling

Jess assay Western blotting analysis was performed to examine the effect of TQ on JAK/STAT signaling in HL60 leukemia cells. The cells were incubated with 2 µM TQ for 48 h; then, the protein was extracted from treated and untreated HL60 leukemia cells for the protein analysis. The results showed that TQ significantly suppressed the expression of JAK2, STAT3, and STAT5 proteins (*p* < 0.001) (Figure 8). The results also revealed that TQ inhibited the activation of JAK/STAT signaling through dephosphorylation of STAT3 and STAT5 proteins.

### 3.6. TQ Inhibited PI3K/AKT Signaling

The effect of TQ treatment was examined on PI3K/AKT protein expression *(*Figure 9). The expression of PI3K and AKT. The results revealed that PI3K and AKT protein expression decreased significantly in the TQ-treated samples (*p* < 0.001). In addition, TQ treatment significantly dephosphorylated PI3K and AKT (*p* < 0.01). Furthermore, the PI3K/AKT negative regulator protein, PTEN, was also detected. TQ increased PTEN protein expression significantly in treated cells (*p* < 0.001).

### 3.7. TQ Inhibited c-Myc Protein Expression

As shown in Figure 10, the effect of TQ on c-Myc protein expression in HL-60 cells was also examined, and the results showed a marked decrease in c-Myc protein after TQ treatment (*** *p* < 0.001).

Based on these results, TQ inhibited the protein expression and phosphorylation of JAK/STAT and PI3K/AKT/mTOR proteins, resulting in the inhibition of c-Myc protein expression.

## 4. Discussion

The c-Myc proto-oncogene is markedly expressed in the HL60 cell [3]. C-Myc has a crucial role in leukemogenesis through inducing proliferation and inhibiting apoptosis in leukemia cells [32]. The overexpression of c-Myc in cancers is correlated with chemotherapy resistance [33]. Drugs cannot directly target the c-Myc gene [6]. Thus, alternative indirect methods to suppress c-Myc function are vitally needed. The JAK/STAT and PI3K/AKT/mTOR pathways are constitutively activated in leukemia leading to c-Myc overexpression [34,35]. Therefore, the current study aimed to determine the effect of TQ on c-Myc by affecting upstream c-Myc regulators: the JAK/STAT and PI3K/AKT/mTOR signaling pathways.

The current study investigated the molecular effect mechanisms of TQ on HL60 AML cells. NGS analysis was performed to study the mRNA changes in leukemia cells treated with 2 µM TQ compared to untreated cells. Fifty-four significantly upregulated genes and 60 significantly upregulated genes associated with TQ treatment were identified in a dose- and time-dependent manner. The 114 affected genes were analyzed by DAVID analysis, which showed that the efficacy of TQ in treating leukemia cells may be related to the biological process of cell proliferation., apoptosis, immune and inflammatory response, and angiogenesis.

The NGS results showed downregulation of JAK/STAT and PI3K/AKT/mTOR genes in TQ-treated cells. The results also revealed significant downregulation in genes that induced the activation of JAK/STAT and PI3K/AKT/mTOR pathways (Figure 11) such as chemokine (C–C motif) ligand 2 (*CCL2*) [36], colony-stimulating factor 1 receptor (*CSF1R*) [37], CXC chemokine receptor 2 (*CXCR2*) [38], interleukin 6 receptor (IL6R) [39], G protein-coupled receptors (GPCRs) [40], and chitinase 3 like 1 protein (Chi3L1) [41]. These findings suggest that TQ induces apoptosis and inhibits cancer cell proliferation by inhibiting the JAK/STAT and PI3K/AKT/mTOR signaling pathways. This suggestion is supported by previous studies where TQ induced apoptosis by inhibiting the activation of the PI3K/AKT pathway in the oral squamous carcinoma KB cell line [42] and inhibited cell proliferation through suppression of the JAK/STAT pathway in human multiple myeloma cells [43]. In contrast, our results showed the upregulation of genes that functioned to inhibit the JAK/STAT and PI3K/AKT/mTOR pathways’ activation including PRDI-BF1 and RIZ homology domain-containing 8 (*PRDM8*) [44] and RAP1 GTPase activating protein (*RAP1GAP*) [45]. In the present study, there was a significant upregulation of *PRDM8* (*p* < 0.044) and RAP1GAP (*p* < 0.029) in TQ-treated HL60 leukemia cells, suggesting that TQ inhibited HL60 cell growth and suppressed c-Myc expression by inhibition of JAK/STAT and PI3K/AKT/mTOR signaling through upregulation of *PRDM8* and *RAP1GAP*.

In order to verify the effect of TQ on the regulation of gene expression in the JAK/STAT and PI3K/AKT/mTOR pathways, the *JAK2*, *STAT3*, *STAT5a*, *STAT5b*, *PI3K*, *AKT*, *mTOR*, and *c-Myc* genes were selected for further gene expression analysis using RT-qPCR. Our results showed that TQ inhibited JAK/STAT and PI3K/AKT/mTOR pathways by downregulating the mRNA expression of targeted genes. Previous reports demonstrated that inhibition of the STAT3/c-Myc was associated with induced apoptosis in esophageal adenocarcinoma cells [46]. Similarly, the current RT-qPCR results showed that downregulated *STAT3* and *c-Myc* in treated HL60 leukemia cells with TQ was associated with a significant increase in cell apoptosis (*p* < 0.001). These findings suggest that TQ induced cell apoptosis in HL60 leukemia cells by inhibiting *STAT3* and *c-Myc* expression. Additionally, inhibition of STAT5/c-Myc significantly reduced the cell proliferation of cervical cancer cells [47]. In agreement, the present study revealed downregulation of *STAT5* and *c-Myc* and inhibition of cell proliferation with apoptosis induction after TQ treatment of HL60 leukemia cells, suggesting that TQ inhibits cell proliferation and induces apoptosis of HL60 leukemia cells by inhibiting the STAT5/c-Myc mediated signaling. These findings are also supported by our previous published studies in which TQ induced downregulation of *JAK2*, *STAT3*, and *STAT5* and a consequent proliferation inhibition and apoptosis induction in MV4-11 AML cells [48] and K562 chronic myeloid leukemia cells [49].

Moreover, PI3K/AKT exerts a specific biological function by upregulating the expression of *c-Myc*, and the PI3K/AKT/c-Myc axis is a valid therapeutic target in treating esophageal squamous cell carcinoma [50]. The use of PI3K/AKT/mTOR inhibitors downregulates *c-Myc* expression and enhances the anti-leukemic effects of the leukemia drug, all-trans retinoic acid (ATRA), in AML cell lines and primary patient samples [51]. Similarly, our findings showed downregulation of *PI3K*, *AKT*, and *mTOR* genes associated with *c-Myc* downregulation and proliferation reduction in TQ-treated HL60 cells.

Upon stimulation, JAK phosphorylates STAT proteins causing STAT dimerization and nucleus translocation, where they bind to DNA and regulate the transcription of target genes such as *c-Myc* [52]. The protein analysis results in the current study showed that TQ treatment reduced the phosphorylation of JAK2, STAT3, and STAT5 proteins. Our findings are consistent with what has previously been reported: that TQ suppressed STAT3 phosphorylation at Tyr705 due to the inhibition of JAK2 activity in gastric cancer [53]. Moreover, TQ treatment reduces the upstream effectors of STAT5, such as KIT and FLT3, leading to dephosphorylate STAT5 in leukemia cells [54]. In addition, our results show that TQ treatment reduces the expression levels of p-PI3K and p-AKT proteins in leukemia cells and induces phosphatase and tensin homolog (PTEN) expression, which is considered protein tyrosine phosphatases in the PI3K/AKT pathway and tumor suppressor [55], suggesting that TQ inhibits the protein synthesis and phosphorylation of the PI3K/AKT pathway through activation of negative regulator PTEN.

The results of the current study showed that TQ inhibited the c-Myc protein expression in HL60 cells. Recent reports have shown that TQ treatment decreases the c-Myc level in AML cells [56]. Additionally, many studies confirm a principal role of c-Myc in proliferation, cell cycle regulation, apoptosis, and differentiation in AML [57], suggesting that TQ reduces the proliferation of AML through inhibiting c-Myc. We hypothesized that TQ-mediated c-Myc downregulation inhibits the upstream regulators: signaling pathways such as JAK/STAT and PI3K/AKT/mTOR. Consistent with the previous report, TQ treatment in this study significantly decreased c-Myc protein expression levels in treated HL60 cells compared with untreated cells. These data suggest that TQ decreases c-Myc protein expression through inhibiting the JAK/STAT and PI3K/AKT/mTOR signaling pathways in HL60 cells.

## 5. Conclusions

Our findings confirm the potential effects of TQ in inhibiting cell proliferation and inducing apoptosis of HL60 leukemia cells through indirectly targeting c-Myc oncogene expression by inhibiting JAK/STAT and PI3K/AKT/mTOR signaling. The findings of the present study nominate the c-Myc as a therapeutic target and TQ as a potential candidate for the treatment of AML patients. However, further studies are required to investigate the effect of TQ on AML in vivo experiments.

## Figures and Tables

**Figure 1 pharmaceuticals-15-00307-f001:**
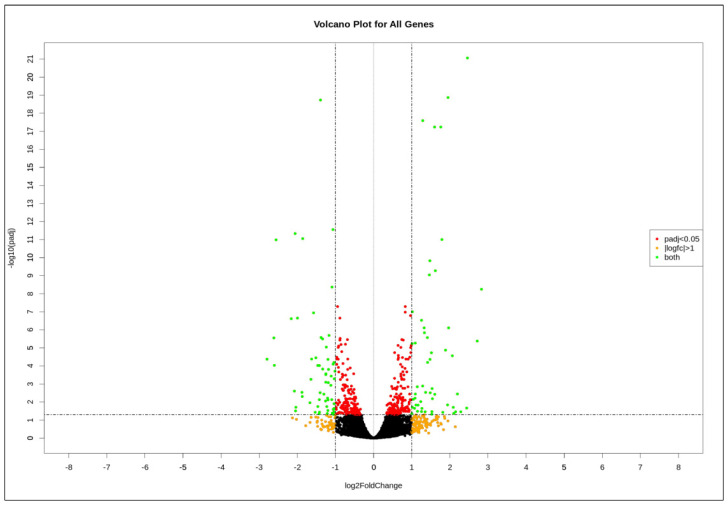
The volcano plot above shows the distribution of all the DEGs identified in this study. Each dot represents a single gene. Green dots are significant genes with *p*−*adj* < 0.01 and |log2 fold change| > 1 (equivalent to a fold change of magnitude greater than 2).

**Figure 2 pharmaceuticals-15-00307-f002:**
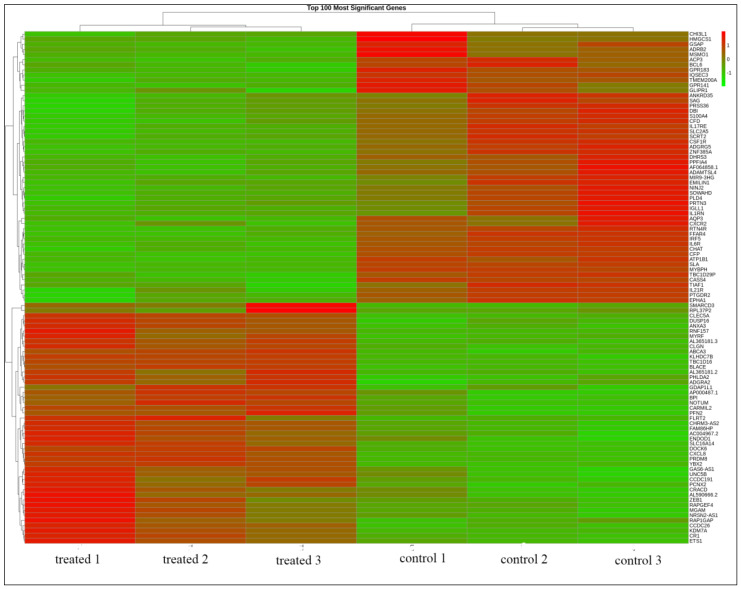
Heatmap of the top 100 significant DEGs in TQ−treated cell samples compared to control samples *(p*−*adj* < 0.01). The red color indicates upregulation, and the green color indicates downregulation. Each row represents one gene.

**Figure 3 pharmaceuticals-15-00307-f003:**
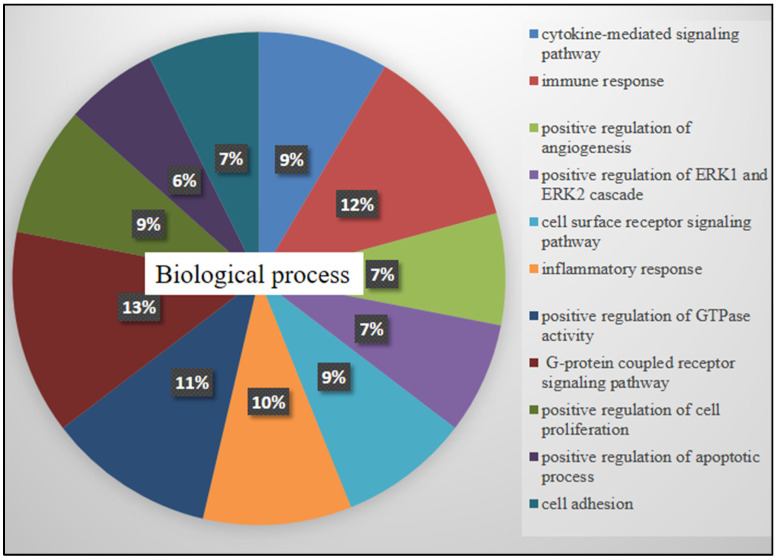
DAVID software analyzed the possible biological processes of the significantly affected genes in treated HL60 cells. Biological activation based on extracellular signals was highly affected after TQ treatment. The area of each category represents the percentages of involved genes to each other (*p* < 0.05).

**Figure 4 pharmaceuticals-15-00307-f004:**
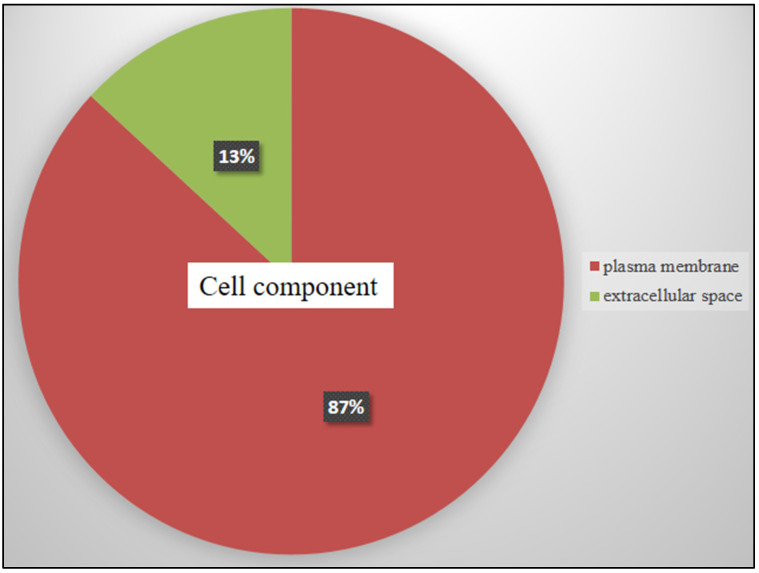
DAVID software analyzed the possible active locations of the significantly affected genes in the cellular components of TQ-treated HL60 cells. The majority of genes were activated in the plasma membrane, which suggests that TQ affects signaling activation in the plasma membrane. The area of each category represents the percentages of involved genes to each other (*p* < 0.05).

**Figure 5 pharmaceuticals-15-00307-f005:**
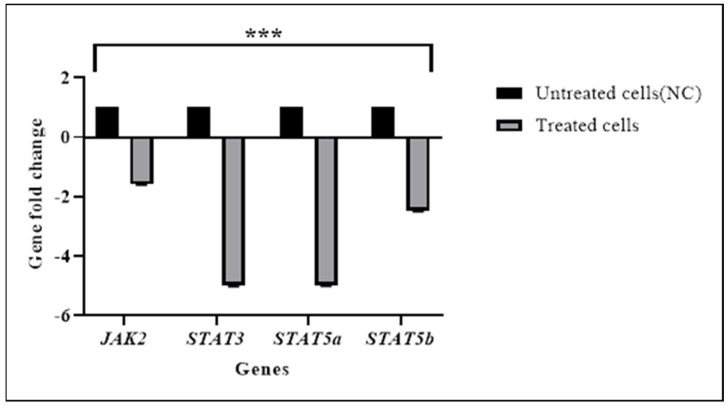
RT−qPCR results of *JAK2*, *STAT3*, *STAT5a*, and *STAT5b* expression in HL60 cells. The relative normalization of RT−qPCR showed that TQ significantly downregulated the expression of targeted genes in treated cells. *STAT3* and *STAT5a* were downregulated in treated cells approximately 5−fold lower compared with untreated cells. Data are presented as the mean ± SEM. *(*** p* < 0.001).

**Figure 6 pharmaceuticals-15-00307-f006:**
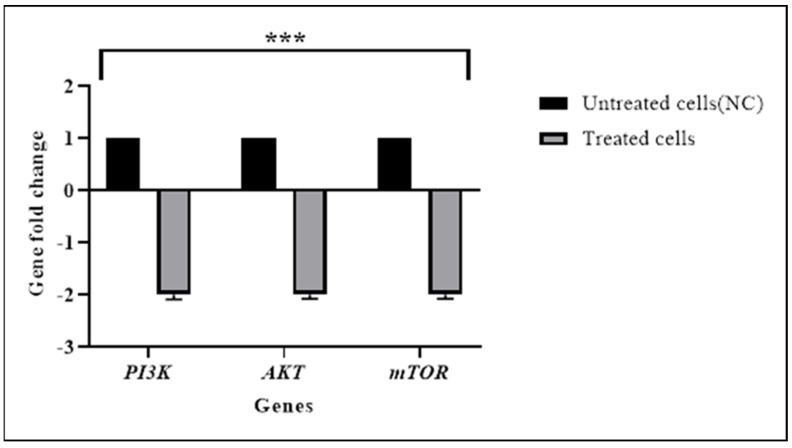
RT−qPCR results of *PI3K*, AKT, and *mTOR* expression in HL60 cells. The relative normalization of RT−qPCR showed that TQ significantly downregulated the expression of the targeted genes in treated cells. *PI3K*, AKT, and *mTOR* were downregulated in treated cells approximately 2-fold lower than in untreated cells. Data are presented as the mean ± SEM (*** *p* < 0.001).

**Figure 7 pharmaceuticals-15-00307-f007:**
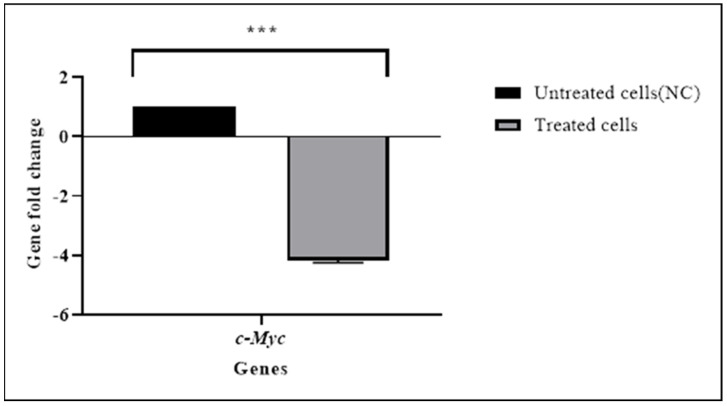
The c−*Myc* expression level was determined using a RT−qPCR assay in HL60 cells. RT−qPCR revealed that TQ significantly downregulated *c*−*Myc* gene expression by 4−fold in treated cells, and the untreated cells were used as a negative control group. Data are presented as the mean ± SEM (*** *p* < 0.001).

**Figure 8 pharmaceuticals-15-00307-f008:**
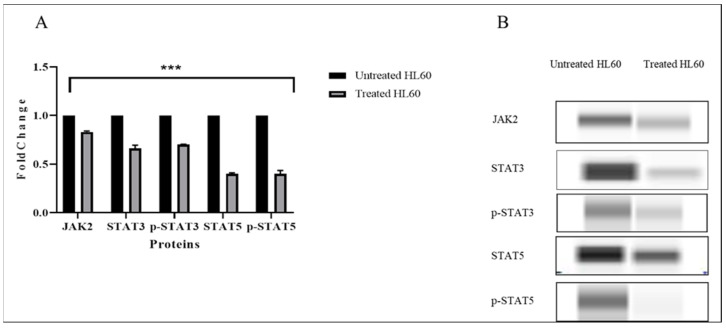
Effects of TQ on the expression of JAK/STAT proteins in HL60 cells. (**A**) JAK2, STAT3, p-STAT3, STAT5, and p-STAT5 protein expression in TQ-treated cells compared with untreated cells or negative control. Data are presented as the mean ± SEM (**** p* < 0.001). (**B**) Images of JAK2, STAT3, p-STAT3, STAT5, and p-STAT5 protein expression from capillary Western blotting in treated and untreated HL60 cells.

**Figure 9 pharmaceuticals-15-00307-f009:**
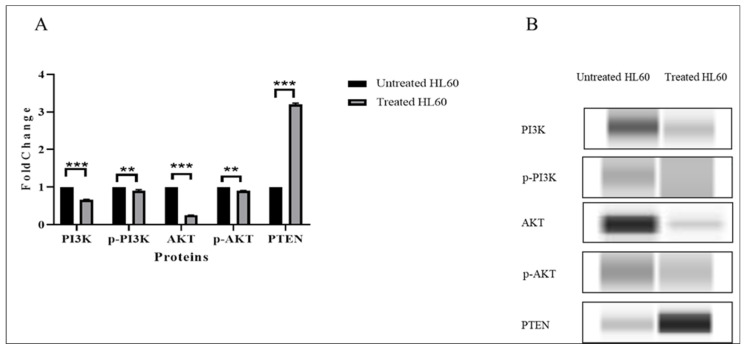
Effects of TQ on the expression of PI3K/AKT proteins in HL60 cells. (**A**) PI3K, p-PI3K, AKT, p-AKT, and PTEN protein expression in TQ-treated cells compared with untreated cells or negative control. Data are presented as the mean ± SEM (*** *p* < 0.001; ** *p* < 0.01). (**B**) Image of PI3K, p-PI3K, AKT, p-AKT, and PTEN protein expression from capillary Western blotting in treated and untreated HL60 cells.

**Figure 10 pharmaceuticals-15-00307-f010:**
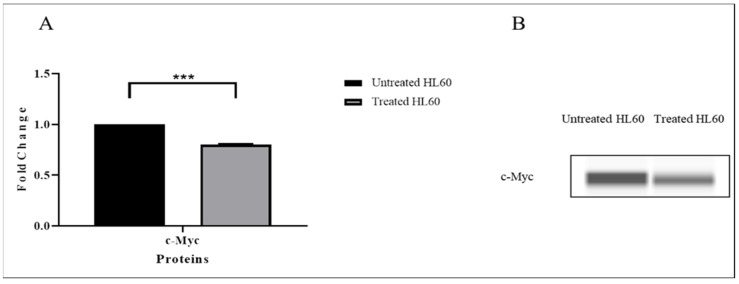
Effects of TQ on the expression of c-Myc protein in HL60 cells. (**A**) c-Myc protein expression in TQ-treated cells compared with untreated cells or negative control. Data are presented as the mean ± SEM (*** *p* < 0.001). (**B**) Image of c-Myc protein expression from capillary Western blotting in treated and untreated HL60 cells.

**Figure 11 pharmaceuticals-15-00307-f011:**
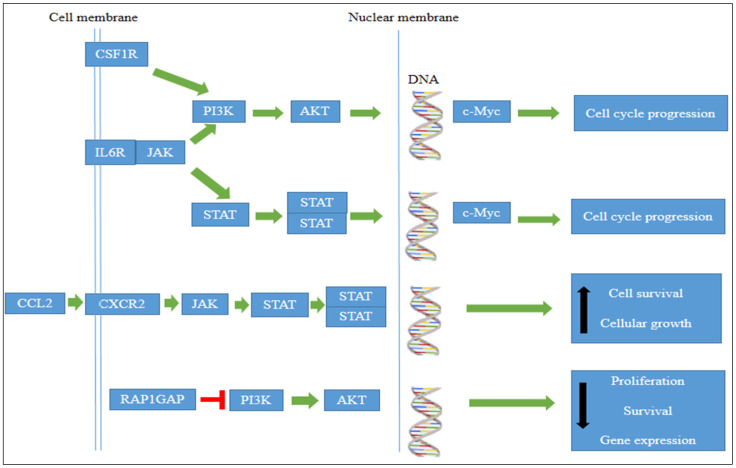
Image showing how significantly expressed genes affect JAK/STAT, PI3K/AKT, and c-Myc activation. Adapted from DAVIVD software.

**Table 1 pharmaceuticals-15-00307-t001:** Sequences of primers used to quantify gene expression by RT-qPCR.

Gene Name	Primer Sequence (5′ to 3′)	References
*JAK2*	F: TGTCTTACCTCTTTGCTCAGTGGCG R: CAATGACATTTTCTCGCTCGACAGC	[24]
*STAT3*	F: GATTGACCAGCAGTATAGCCGCTTC R: CTGCAGTCTGTAGAAGGCGTG	[25]
*STAT5a*	F: GTCCTGAAGACCCAGACCAA R: GTTGCGGGTGTTCTCATTTT	[26]
*STAT5b*	F: CATTTTCCCATTGAGGTGCG R: GGGTGGCCTTAATGTTCTCC	[27]
*PI3K*	F: TTAGCTATTCCCACGCAGGA R: CACAATAGTGTCTGTGACTC	[28]
*AKT*	F: CTGAGATTGTGTCAGCCCTG R: CACAGCCCGAAGTCTGTGATCTTA	[29]
*mTOR*	F: ATGCAGCTGTCCTGGTTCTC R: AATCAGACAGGCACGAAG	[28]
*c-Myc*	F: CCACAGCAAACCTCCTCACA R: TCCAACTTGACCCTCTTGGC	[30]
*β-actin*	F: CTGGCACCCAGGACAATG R: GCCGATCCACACGGAGTA	[31]

**Table 2 pharmaceuticals-15-00307-t002:** Biological process and molecular function of upregulated genes effected by TQ treatment.

Category	Term	Similarity Score	*p*-Value	Genes
GOTERM_BP_DIRECT	Positive regulation of endothelial cell migration	1.00	0.0037	*ETS1*, *ADGRA2*, and *ANXA3*
GOTERM_BP_DIRECT	Regulation of angiogenesis	0.79	0.0037	*ETS1*, *ADGRA2*, and *ANXA3*
GOTERM_BP_DIRECT	Positive regulation of angiogenesis	1.00	0.021	*CXCL8*, *ETS1*, and *ANXA3*
KEGG_PATHWAY	Pathways in cancer	0.79	0.021	*CXCL8*, *ETS1*, and *ANXA3*
GOTERM_BP_DIRECT	Oligodendrocyte development	1.00	0.044	*PRDM8* and *MYRF*
GOTERM_BP_DIRECT	Immune response	1.00	0.049	*CXCL8*, *ETS1*, and *BPI*
GOTERM_BP_DIRECT	Negative regulation of cell proliferation	0.85	0.049	*CXCL8*, *ETS1*, and *BPI*
GOTERM_MF_DIRECT	Ras GTPase binding	1.00	0.029	*RAP1GAP* and *RAPGEF4*
KEGG_PATHWAY	Rap1 signaling pathway	0.79	0.029	*RAP1GAP* and *RAPGEF4*

**Table 3 pharmaceuticals-15-00307-t003:** Biological process and molecular function of downregulated genes effected by TQ treatment.

Category	Term	Similarity Score	*p*-Value	Genes
GOTERM_BP_DIRECT	Cytokine-mediated signaling pathway	1.00	0.000047	*CCL2*, *CSF1R*, *IRF5*, *IL17RE*, *IL6R*, and *RTN4R*
GOTERM_BP_DIRECT	Positive regulation of ERK1 and ERK2 cascade	1.00	0.00019	*CCL2*, *GPR183*, *GPR55*, *CHI3L1*, *CSF1R*, and *FFAR4*
GOTERM_BP_DIRECT	Inflammatory response	1.00	0.00097	*BCL6*, *CCL2*, *CXCR2*, *GPR68*, *CHI3L1*, *CSF1R*, and *IL17RE*
UP_KEYWORDS	Inflammatory response	0.81	0.00097	*BCL6*, *CCL2*, *CXCR2*, *GPR68*, *CHI3L1*, *CSF1R*, and *IL17RE*
GOTERM_BP_DIRECT	Cell surface receptor signaling pathway	1.00	0.0014	*CCL2*, *CXCR2*, *EPHA1*, *SAG*, *ADGRG5*, and *ADRB2*
GOTERM_BP_DIRECT	Immune response	1.00	0.0088	*CCL2*, *GPR183*, *CFP*, *IGLL1*, *IL1RN*, and *PTGDR2*
GOTERM_BP_DIRECT	Cell adhesion	1.00	0.012	*ATP1B1*, *CCL2*, *CASS4*, *EMILIN1*, *MYBPH*, and *NINJ2*
UP_KEYWORDS	Cell adhesion	0.81	0.012	*ATP1B1*, *CCL2*, *CASS4*, *EMILIN1*, *MYBPH*, and *NINJ2*
GOTERM_BP_DIRECT	Positive regulation of apoptotic process	1.00	0.013	*ADAMTSL4*, *BCL6*, *ARHGEF3*, *S100B*, and *IRF5*
GOTERM_BP_DIRECT	Positive regulation of cell proliferation	1.00	0.013	*CXCR2*, *EPHA1*, *S100B*, *CSF1R*, *IL6R*, and *PRTN3*
GOTERM_BP_DIRECT	Cellular response to macrophage colony-stimulating factor stimulus	1.00	0.018	*CCL2* and *CSF1R*
GOTERM_BP_DIRECT	Cell shape regulation	0.79	0.018	*CCL2* and *CSF1R*
GOTERM_BP_DIRECT	G protein-coupled receptor pathway	1.00	0.018	*CCL2*, *GPR141*, *GPR183*, *GPR55*, *GPR68*, *ADGRG5*, *FFAR4*, and *PTGDR2*
GOTERM_MF_DIRECT	G protein-coupled receptor activity	0.92	0.018	*CCL2*, *GPR141*, *GPR183*, *GPR55*, *GPR68*, *ADGRG5*, *FFAR4*, and *PTGDR2*
UP_KEYWORDS	Transducer	0.79	0.018	*CCL2*, *GPR141*, *GPR183*, *GPR55*, *GPR68*, *ADGRG5*, *FFAR4*, and *PTGDR2*
UP_KEYWORDS	G protein-coupled receptor	0.79	0.018	*CCL2*, *GPR141*, *GPR183*, *GPR55*, *GPR68*, *ADGRG5*, *FFAR4*, and *PTGDR2*
GOTERM_BP_DIRECT	Relaxation of cardiac muscle	1.00	0.036	*ATP1B1* and *RGS2*
KEGG_PATHWAY	cGMP–PKG signaling pathway	0.79	0.036	*ATP1B1* and *RGS2*
GOTERM_BP_DIRECT	Protein O-linked fucosylation	1.00	0.036	*ADAMTSL4* and *CFP*
GOTERM_BP_DIRECT	Complement activation, alternative pathway	1.00	0.039	*CFD* and *CFP*
GOTERM_BP_DIRECT	Positive regulation of angiogenesis	1.00	0.048	*CXCR2*, *EPHA1*, and *CHI3L1*

**Table 4 pharmaceuticals-15-00307-t004:** The fold change of JAK/STAT and genes in NGS and RT−qPCR.

Genes	Fold Change		
	NGS	*p*-Value	RT-qPCR
*JAK2*	–0.52	0.006	−1.6
*STAT3*	−0.31	0.012	−5.0
*STAT5a*	−0.31	0.025	−5.0
*STAT5b*	−0.49	0.001	−2.5

**Table 5 pharmaceuticals-15-00307-t005:** The fold change of PI3K/AKT/mTOR genes in NGS and RT-qPCR.

Genes	Fold Change		
	NGS	*p*-Value	RT-qPCR
*AKT*	−0.89	0.006	−2.0
*PI3K*	−0.59	0.003	−2.0
*mTOR*	−0.26	0.043	−2.0

**Table 6 pharmaceuticals-15-00307-t006:** The fold change of the *c-Myc* gene in NGS and RT-qPCR.

Genes	Fold Change		
	NGS	*p*-Value	RT-qPCR
*c-Myc*	−0.625	0.00001	−4.16

## Data Availability

Data is contained within the article.

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
