# Peer review of "Gene Expression Profiling and Protein Analysis Reveal Suppression of the C-Myc Oncogene and Inhibition JAK/STAT and PI3K/AKT/mTOR Signaling by Thymoquinone in Acute Myeloid Leukemia Cells"

_pharmaceuticals, 2022, doi:10.3390/ph15030307_

Round 1
Reviewer 1 Report
In this research paper, the authors investigated the effects of one of the popular natural products thymoquinone (TQ) on the expression of different genes in a cell line of acute leukaemia. The authors figured out the effects of TQ on the expression profiles of mRNA first and then on some proteins involved in cellular signalling pathways. The authors made a conclusion of the anticancer effect of TQ which can trigger cell apoptosis. There are several conceptual issues related to the experimental design of this study and overall, the low quality of the text. It can be listed as follows:
- The authors have tested only one concentration of TQ. What was the reason to choose such a concentration? There is no explanation for this. Also, no references were given to other studies where it might have been tested.
- Further to different concentrations of TQ, it must be also tested the time-dependence of its effects. Why did the authors incubate TQ for 48 h? The reason for this should be made clear and the authors have to confirm that this treatment is optimal.
- The conclusions of the anticancer effect of TQ are based on testing only one cell line, HL-60 cells. At least another cell line should be tested in order to argue for a versatile anticancer effect of TQ on acute leukaemia.
- The authors used only three repeats of H-L60 leukaemia cells. While n = 3 represents minimum replicates required for testing biological variability, these cells must be obtained from different preparations. Were the three repeats in this study from the same passage?
- Please add the composition of the culture medium into the methods section while describing cell culturing in the RPMI medium and give references for its use by others.
- What do the authors mean by “integral component” (page 7)? Please list such components for clarity. What are the differences between the “integral component of plasma membrane” and “integral component of membrane” and “plasma membrane” (Fig.4)?
- Please show values for all sections in Figures 3 and 4 to clarify their proportions into the whole bar chat.
- There is no reference in the text for Fig. 5.
- All results sections contain very little information about why one or another gene or protein of interest were tested and what the obtained results mean. By scientific style, it is expected that authors give a conсise introduction to each results section on this and make a short conclusion what the particular set of obtained results mean.
- The results would benefit if sections are combined for the changed expression of genes and the related signalling pathways. For example:
- TQ inhibits the JAK/STAT signalling pathways by downregulating the mRNA and protein expression levels.
- The effects of TQ on the gene expression and c-Myc protein level.
- The text needs extensive editing for the style and language – there are numerous grammar mistakes, with many unfinished sentences (e.g. lines 459-460, 480-481), non-appropriate article use, errors, etc.
Author Response
Date:16 February 2022
Dear Editor
Thank you for giving us the opportunity to submit the revised manuscript entitled “Gene Expression Profiling and Protein Analysis Reveal Inhibition of the c-MYC Oncogene by Thymoquinone through Suppression of Signaling Pathways in Acute Myeloid Leukemia Cells” for publication in Pharmaceuticals Journal.
We appreciate your and reviewers’ valuable comments on the manuscript. Please be informed that we have intensively revised the manuscript and highlighted the changes in the manuscript track changes.
Reviewer 1
Comments and Suggestions for Authors
In this research paper, the authors investigated the effects of one of the popular natural products thymoquinone (TQ) on the expression of different genes in a cell line of acute leukaemia. The authors figured out the effects of TQ on the expression profiles of mRNA first and then on some proteins involved in cellular signalling pathways. The authors made a conclusion of the anticancer effect of TQ which can trigger cell apoptosis. There are several conceptual issues related to the experimental design of this study and overall, the low quality of the text. It can be listed as follows:
- The authors have tested only one concentration of TQ. What was the reason to choose such a concentration? There is no explanation for this. Also, no references were given to other studies where it might have been tested.
Author Response
Thank you for the comment and with all respect to the reviewer, I have mentioned in the manuscript that the IC50 of TQ on HL60 cells was 2 μM at 48 h according to the MTT assay in our published data. Please refer to “method of the manuscript under TQ treatment” line 100.
- Further to different concentrations of TQ, it must be also tested the time-dependence of its effects. Why did the authors incubate TQ for 48 h? The reason for this should be made clear and the authors have to confirm that this treatment is optimal.
Author Response
Yes it was after optimization and determination of IC50, Please refer to the prvious publish Data which has been cited for this point: 10.31557/APJCP.2021.22.3.879
- The conclusions of the anticancer effect of TQ are based on testing only one cell line, HL-60 cells. At least another cell line should be tested in order to argue for a versatile anticancer effect of TQ on acute leukaemia.
Author Response
Thank you for the comment and please consider that we have studied the effects of TQ on three types of leukemia cell lines K562 (CML cells), MV4-11, and HL60 (AML cells). The effect of TQ on MV4-11 and K562 cells were published in pharmaceuticals Journal (48), and Asian Pacific Journal of Cancer Prevention (49), respectively.
- The authors used only three repeats of H-L60 leukaemia cells. While n = 3 represents minimum replicates required for testing biological variability, these cells must be obtained from different preparations. Were the three repeats in this study from the same passage?
Author Response
Thanks to the reviewer and please consider that once we talking about cell line, we are dealing with millions of cells and NOTonly three samples. Yes the protein and RNA have been extracted from the same passage.
- Please add the composition of the culture medium into the methods section while describing cell culturing in the RPMI medium and give references for its use by others.
Author Response
Thanks for the reviewer and please be informed that the contents of media have been mentioned in method of the manuscript under “Cell line and growth media”
- What do the authors mean by “integral component” (page 7)? Please list such components for clarity. What are the differences between the “integral component of plasma membrane” and “integral component of membrane” and “plasma membrane” (Fig.4)?
Author Response
Thanks a lot to the reviewer and the comment has considered and the Figure 4 has been modified in the revised manuscript.
- Please show values for all sections in Figures 3 and 4 to clarify their proportions into the whole bar chat.
Author Response
Thank you, the comment has been considered and the percentage of each part has been added to the chart in the revised manuscript.
- There is no reference in the text for Fig. 5.
Author Response
The comment has considered and Figure 5 has been cited in the text of revised manuscript, please refer to results under “TQ downregulated the expression of JAK/STAT pathways genes”.
- All results sections contain very little information about why one or another gene or protein of interest were tested and what the obtained results mean. By scientific style, it is expected that authors give a concise introduction to each results section on this and make a short conclusion what the particular set of obtained results mean.
Author Response
Thanks for the comment and I think it is about Tables 2 & 3 in the Results section. The comment has considered and the statement has been added in the revised manuscript “DAVID analysis of the NGS results showed that the differentially expressed genes were primarily involved in signaling pathways, proliferation, angiogenesis, and apoptosis processes in leukemia cells. Some of these genes control the activation of the JAK/STAT and PI3K/AKT/mTOR signaling such as CCL2, CXCR2, and PRDM8”.
- The results would benefit if sections are combined for the changed expression of genes and the related signalling pathways. For example:
- TQ inhibits the JAK/STAT signalling pathways by downregulating the mRNA and protein expression levels.
- The effects of TQ on the gene expression and c-Myc protein level.
Author Response
Thanks for highlighting this point of view, the following statements have been added to discussion in line 434: “Our results showed that TQ inhibited JAK/STAT and PI3K/AKT/mTOR pathways by downregulating the mRNA expression of targeted genes). Also in Line 474: “The results of the current study showed that TQ inhibited the c-Myc protein expression in HL60 cells”
The text needs extensive editing for the style and language – there are numerous grammar mistakes, with many unfinished sentences (e.g. lines 459-460, 480-481), non-appropriate
Author Response
The revised manuscript has edited in MDPI English editing services, please refer to the attached editing certificate.

Reviewer 2 Report
Authors of the manuscript compared expression profiles (RNA-seq, RT-qPCR) in HL60 cells treated or non-treated with Thymoquinone (TQ). Manuscript only compares the transcription profiles of differentially expressed genes with an intention to find mechanism of TQ effect on cell proliferation. However, authors are making several overstatements.
Authors mention the inhibitory effect of TQ on cell proliferation, but do not show any relevant data. Even though it is known the TQ effect on cell proliferation should be presented in the manuscript to prove that the expected effect is exerted in author's setting.
Authors claim that c-myc transcription is inhibited by TQ through suppression of JAK/STAT and PI3K/AKT/mTOR pathways. It is probable but can't be concluded from the data. The data show only correlation indicating possible causation, but do not provide any proof that c-myc is inhibited by TQ through the pathways. The impact of the data would be improved if at least a competitive experiment combining TQ and some of the known well characterized JAK/STAT activators would be performed. If TQ would be able to at least partially restrain the activator effect it says something about the mechanism. Unfortunately, the data in the current form have very low scientific significance.
Author Response
Date:16 February 2022
Dear Editor
Thank you for giving us the opportunity to submit the revised manuscript entitled “Gene Expression Profiling and Protein Analysis Reveal Inhibition of the c-MYC Oncogene by Thymoquinone through Suppression of Signaling Pathways in Acute Myeloid Leukemia Cells” for publication in Pharmaceuticals Journal.
We appreciate your and reviewers’ valuable comments on the manuscript. Please be informed that we have intensively revised the manuscript and highlighted the changes in the manuscript track changes.
Reviewer 2
Comments and Suggestions for Authors
Authors of the manuscript compared expression profiles (RNA-seq, RT-qPCR) in HL60 cells treated or non-treated with Thymoquinone (TQ). Manuscript only compares the transcription profiles of differentially expressed genes with an intention to find mechanism of TQ effect on cell proliferation. However, authors are making several overstatements.
Author Response
Thanks for the comment, the NGS analysis showed 114 significant differentially expressed genes after TQ treatment. Many of them are directly implicated in regulation of JAK/STAT and PI3K/AKT pathways. So, we have tracked the effect of the TQ-treatment on the signaling pathways and conformed by using RT-qPCR. In addition, protein analysis using Jess assay Western blotting analysis.
Authors mention the inhibitory effect of TQ on cell proliferation, but do not show any relevant data. Even though it is known the TQ effect on cell proliferation should be presented in the manuscript to prove that the expected effect is exerted in author's setting.
Author Response
Dear reviewer, we have used the IC50 based on our published data which has been cited (14) in the current manuscript.
Authors claim that c-myc transcription is inhibited by TQ through suppression of JAK/STAT and PI3K/AKT/mTOR pathways. It is probable but can't be concluded from the data. The data show only correlation indicating possible causation, but do not provide any proof that c-myc is inhibited by TQ through the pathways. The impact of the data would be improved if at least a competitive experiment combining TQ and some of the known well characterized JAK/STAT activators would be performed. If TQ would be able to at least partially restrain the activator effect it says something about the mechanism. Unfortunately, the data in the current form have very low scientific significance.
Author Response
It is known that direct targeting of c-MYC oncogene is impossible, and it could be targeted indirectly through mediated signaling pathways such as JAK/STAT and PI3K/AKT/mTOR. For this reason, we hypothesised that inhibition of JAK/STAT and PI3K/AKT/mTOR signaling could suppress c-MYC expression. The results revealed that TQ significantly inhibited JAK/STAT and PI3K/AKT signaling with markedly down-regulation of c-MYC oncoprotein.

Reviewer 3 Report
Dear Authors,
This is a good paper analyzing the effects of Thymoquinone on HL-60. The experiments were appropriate for the goal of the study. I only have some questions and comments:
- Why did you choose 48 hours as incubation period?
- For mRNA expression profiling, which kit did you use? Whole RNAseq, targeted RNAseq? It is important to specify if you analyzed the whole transcriptome or only targeted one.
- On the discussion, at the end, I would include future directions, what are the next steps of this work?
Thank you very much.
Author Response
Date:16 February 2022
Dear Editor
Thank you for giving us the opportunity to submit the revised manuscript entitled “Gene Expression Profiling and Protein Analysis Reveal Inhibition of the c-MYC Oncogene by Thymoquinone through Suppression of Signaling Pathways in Acute Myeloid Leukemia Cells” for publication in Pharmaceuticals Journal.
We appreciate your and reviewers’ valuable comments on the manuscript. Please be informed that we have intensively revised the manuscript and highlighted the changes in the manuscript track changes.
Reviewer 3
Comments and Suggestions for Authors
Dear Authors,
This is a good paper analyzing the effects of Thymoquinone on HL-60. The experiments were appropriate for the goal of the study. I only have some questions and comments:
- Why did you choose 48 hours as incubation period?
Author Response
Thank you very much for the comment. We have used the IC50 according to our published data which has been cited (14) in the current manuscript.
- For mRNA expression profiling, which kit did you use? Whole RNAseq, targeted RNAseq? It is important to specify if you analyzed the whole transcriptome or only targeted one.
Author Response
We have analysed the whole transcriptome for gene expression profiling, so the comment has considered and the statement has been added in the beginning part of results in the revised manuscript.
- On the discussion, at the end, I would include future directions, what are the next steps of this work?
Author Response
Thanks for the comment and the statement was added in revised manuscript at the end of conclusion “However, further studies are required to investigate the effect of TQ on AML in-vivo experiments”.

Reviewer 4 Report
The manuscript entitled “Gene expression profiling and protein analysis reveal inhibition of c-myc oncogene by Thymoquinone through suppression of signaling pathways in acute myeloid cells” by Almajali B et al. submitted to Pharmaceuticals journal. The authors investigate the effects of Thymoquinone on the activation of upstream regulators of c-Myc, JAK/STAT and PI3K/AKT/mTOR pathways in HL60 leukemia cells. The research was appropriate designed and the methods were adequate to respond the questions. The results were properly described and support the conclusions. In my opinion the manuscript should be publish in Pharmaceuticals.
Author Response
Reviewer 4
Comments and Suggestions for Authors
The manuscript entitled “Gene expression profiling and protein analysis reveal inhibition of c-myc oncogene by Thymoquinone through suppression of signaling pathways in acute myeloid cells” by Almajali B et al. submitted to Pharmaceuticals journal. The authors investigate the effects of Thymoquinone on the activation of upstream regulators of c-Myc, JAK/STAT and PI3K/AKT/mTOR pathways in HL60 leukemia cells. The research was appropriate designed and the methods were adequate to respond the questions. The results were properly described and support the conclusions. In my opinion the manuscript should be publish in Pharmaceuticals.
Author Response
Thank you very much to the reviewer for the kind comments.

Round 2
Reviewer 1 Report
None.
Author Response
Date: February 18, 2022
REVIEWER 1
Comments and Suggestions for Authors: None.
Thank you very much to the reviewer for accepting all responses to the comments.

Reviewer 2 Report
I am not satisfied with the authors response:
Author Response
It is known that direct targeting of c-MYC oncogene is impossible, and it could be targeted indirectly through mediated signaling pathways such as JAK/STAT and PI3K/AKT/mTOR. For this reason, we hypothesised that inhibition of JAK/STAT and PI3K/AKT/mTOR signaling could suppress c-MYC expression. The results revealed that TQ significantly inhibited JAK/STAT and PI3K/AKT signaling with markedly down-regulation of c-MYC oncoprotein.
It doesn't respond to my comment. If authors are not willing to add any additional experiments or data they should at least change the title of the manuscript. The title indicates that the manuscript provides proof that c-myc is suppressed through the signaling pathways, but it provides only correlations. The title should clearly say that the data only indicates such causality.
Author Response
Date:February 18, 2022
Dear Editor
Thank you for giving us the opportunity to submit the revised manuscript entitled “Gene Expression Profiling and Protein Analysis Reveal Inhibition of the c-MYC Oncogene by Thymoquinone through Suppression of Signaling Pathways in Acute Myeloid Leukemia Cells” for publication in Pharmaceuticals Journal.
We appreciate your and reviewer’s valuable comment on the manuscript. Please be informed that the TITLE has modified according to the review’s suggestion and the changes have been highlighted with red color at the end of author’s response below.
Reviewer 2
Comments and Suggestions for Authors
I am not satisfied with the authors response:
Author Response
It is known that direct targeting of c-MYC oncogene is impossible, and it could be targeted indirectly through mediated signaling pathways such as JAK/STAT and PI3K/AKT/mTOR. For this reason, we hypothesised that inhibition of JAK/STAT and PI3K/AKT/mTOR signaling could suppress c-MYC expression. The results revealed that TQ significantly inhibited JAK/STAT and PI3K/AKT signaling with markedly down-regulation of c-MYC oncoprotein.
It doesn't respond to my comment. If authors are not willing to add any additional experiments or data they should at least change the title of the manuscript. The title indicates that the manuscript provides proof that c-myc is suppressed through the signaling pathways, but it provides only correlations. The title should clearly say that the data only indicates such causality.
Author’s Response:
Thank you very much to the reviewer and please be informed that the comment has been considered and the title of the manuscript has been modified according to the suggestion, Please refer to the revised manuscript 2.
“GENE EXPRESSION PROFILING AND PROTEIN ANALYSIS REVEAL SUPPRESSION OF THE C-MYC ONCOGENE AND INHIBITION JAK/STAT AND PI3K/AKT/mTOR SIGNALING BY THYMOQUINONE IN ACUTE MYELOID LEUKEMIA CELLS”
